# Humoral Immune Response to CoronaVac in Turkish Adults

**DOI:** 10.3390/vaccines11020216

**Published:** 2023-01-18

**Authors:** Yasemin Cosgun, Nergis Emanet, Ayten Öz Kamiloglu, Evelin Grage-Griebenow, Susann Hohensee, Sandra Saschenbrecker, Katja Steinhagen, Gulay Korukluoglu

**Affiliations:** 1National Arboviruses and Viral Zoonotic Diseases Laboratory, Microbiology Reference Laboratories Department, Public Health General Directorate of Turkey, Ankara 06100, Turkey; 2Virology Unit, Department of Medical Microbiology, Faculty of Medicine, Hacettepe University, Ankara 06230, Turkey; 3EUROIMMUN Turkey, Istanbul 34415, Turkey; 4Institute for Experimental Immunology, EUROIMMUN Medizinische Labordiagnostika AG, 23560 Lübeck, Germany

**Keywords:** CoronaVac, COVID-19, nucleocapsid protein, SARS-CoV-2, spike protein

## Abstract

While most approved vaccines are based on the viral spike protein or its immunogenic regions, inactivated whole-virion vaccines (e.g., CoronaVac) contain additional antigens that may enhance protection. This study analyzes short-term humoral responses against the SARS-CoV-2 spike (S1) and nucleocapsid (NCP) protein in 50 Turkish adults without previous SARS-CoV-2 infection after CoronaVac immunization. Samples were collected before vaccination (t0), 28–29 days after the first vaccine dose and prior to the second dose (t1), as well as 14–15 days after the second dose (t2). Anti-S1 IgG and IgA as well as anti-NCP IgG were quantified using ELISA. At t1, seroconversion rates for anti-S1 IgG, anti-S1 IgA and anti-NCP IgG were 30.0%, 28.0% and 4.0%, respectively, increasing significantly to 98.0%, 78.0% and 40.0% at t2. The anti-NCP IgG median (t2) was below the positivity cut-off, while anti-S1 IgG and IgA medians were positive. Anti-S1 IgG levels strongly correlated with anti-S1 IgA (r_s_ = 0.767, *p* < 0.001) and anti-NCP IgG (r_s_ = 0.683, *p* < 0.001). In conclusion, two CoronaVac doses induced significant increases in antibodies against S1 and NCP. Despite strong correlations between the antibody concentrations, the median levels and seroconversion rates of S1-specific responses exceed those of NCP-specific responses as early as two weeks after the second vaccine dose.

## 1. Introduction

Since 2020, various COVID-19 vaccines have been developed to cope with the spread of severe acute respiratory coronavirus type 2 (SARS-CoV-2). These vaccines are based on distinct antigen platforms (e.g., mRNA, nonreplicating viral vectors, recombinant viral protein, inactivated whole virus), which elicit immune responses that potently neutralize SARS-CoV-2, thereby mediating protective immunity and efficiently reducing the risk of severe disease courses and death [1,2].

The inactivated CoronaVac vaccine (Sinovac Life Sciences, Beijing, China) obtained emergency use authorization in some of the world’s most populated countries (e.g., Brazil, Turkey, China) and accounts for almost a quarter of the COVID-19 vaccine doses delivered globally [3]. Clinical trials in adults demonstrated CoronaVac’s immunogenicity, efficacy at preventing severe COVID-19, and a low potential for side effects [4,5,6,7]. In addition, the effectiveness of CoronaVac to protect against major complications and hospitalization was demonstrated in adults aged ≥70 years [8] and in children [9,10]. This is particularly relevant as COVID-19 is associated with a disproportionately high morbidity and mortality risk among elderly people [11] and may cause multisystem inflammatory syndrome or long-COVID in children [12]. In the elderly, vaccine effectiveness and neutralizing antibody responses against SARS-CoV-2 decline with increasing age, possibly due to senescence of the immune system, suggesting the high relevance of booster immunization in this group [8,13]. Primary immunization with CoronaVac is based on a two-dose schedule that leads to peak antibody levels 20–30 days after the second inoculation, followed by waning immunity [14,15,16]. A third homologous or heterologous dose significantly boosts antibody levels and cellular responses [14,17,18,19]. Like other inactivated virus vaccines, CoronaVac is stable at standard refrigeration temperature, offering advantages for distribution to less developed areas with limited cooling facilities. Containing the full antigenic repertoire of SARS-CoV-2, CoronaVac is likely to elicit a broader immune response than spike-based vaccines. This may confer benefits in reducing the escape from vaccine-induced immunity of emerging SARS-CoV-2 variants of concern [20].

Here, we analyzed and compared short-term humoral responses to the SARS-CoV-2 spike (S1 subunit) and nucleocapsid protein (NCP) in Turkish adults following CoronaVac vaccination.

## 2. Materials and Methods

### 2.1. Samples

In this prospective cohort study, 50 Turkish adults (21 males, 29 females) aged 45.6 ± 7.3 years (range: 27–58 years, median: 46.5 years) without previously documented SARS-CoV-2 infection were vaccinated with two doses of CoronaVac (Sinovac Life Sciences, Beijing, China), administered four weeks apart. Blood samples were collected 1 day before vaccination (t0), 28–29 days after the first vaccine dose and prior to the second dose (t1), and 14–15 days after the second dose (t2). The study was approved by the institutional authority at the Public Health General Directorate of Turkey, and written informed consent was obtained from all participants.

### 2.2. Enzyme-Linked Immunosorbent Assays

Antibodies against S1 and NCP were measured using the Anti-SARS-CoV-2 QuantiVac ELISA IgG, Anti-SARS-CoV-2 ELISA IgA and Anti-SARS-CoV-2 NCP ELISA IgG (EUROIMMUN Medizinische Labordiagnostika AG, Lübeck, Germany). Assays were performed and evaluated according to the manufacturer’s instructions. For anti-S1 IgG, results <25.6 binding antibody units (BAU)/mL were considered negative, ≥25.6 to <35.2 borderline, and ≥35.2 positive. For anti-S1 IgA and anti-NCP IgG, ratios <0.8 considered negative, ≥0.8 to <1.1 borderline, and ≥1.1 positive.

### 2.3. Statistical Analysis

As all data series were not normally distributed (Shapiro–Wilk test), non-parametric statistical testing was applied, with *p*-values < 0.05 indicating significant differences. The Wilcoxon signed-rank test was used to assess the significance of changes in antibody levels in paired samples between different time points. Differences between two independent groups were analyzed by Mann–Whitney U test. Correlation between antibody levels was evaluated using Spearman’s test. Statistical analyses were conducted using SigmaPlot 13.0 (SSI, San Jose, CA, USA) and GraphPad QuickCalcs (GraphPad Software, Inc., San Diego, CA, USA).

## 3. Results

The median anti-S1 IgG levels significantly increased (*p* < 0.001) from <3.2 BAU/mL (interquartile range [IQR] <3.2–3.9) at t0 to 13.8 BAU/mL (IQR 8.8–32.1) at t1, followed by a further increase to 206.7 BAU/mL (IQR 124.1–384.0) at t2 (Figure 1A). Anti-S1 IgA levels also significantly increased (*p* < 0.001) from a median ratio of 0.4 (IQR 0.2–0.5) at t0 to 0.5 (IQR 0.3–0.9) at t1, subsequently rising to a value of 2.0 (IQR 0.9–3.4) at t2 (Figure 1B). The median anti-NCP IgG values remained stable (*p* > 0.05) at a ratio of 0.1 (IQR 0.1–0.2) between t0 and t1. After the second dose, anti-NCP IgG significantly increased (*p* < 0.001) to a median ratio of 0.6 (IQR 0.3–1.1) at t2, which is below the cut-off for reactivity (Figure 1C).

Anti-S1 IgG levels strongly correlated with both anti-S1 IgA (r_s_ = 0.767, *p* < 0.001, Figure 2A) and anti-NCP IgG (r_s_ = 0.624, *p* < 0.001, Figure 2B). A moderate correlation was found between anti-S1 IgA and anti-NCP IgG (r_s_ = 0.527, *p* < 0.001, Figure 2C).

At t1 and t2, there were no statistically significant differences (*p* > 0.05) between the two sexes in the responses against S1 and NCP (Appendix A).

At baseline, all participants were negative for the tested antibodies, apart from two cases (4.0%, 2/50) with anti-S1 IgA borderline results. At t1, seroconversion rates for anti-S1 IgG, anti-S1 IgA and anti-NCP IgG were 30.0% (15/50), 28.0% (14/50) and 4.0% (2/50), respectively. After the second dose, these rates grew further, reaching 98.0% (49/50), 78.0% (39/50) and 40.0% (20/50) at t2, respectively (Table 1).

## 4. Discussion

As the adaptive immune response is a decisive factor for the outcome of SARS-CoV-2 infections, vaccinations are crucial to control the COVID-19 pandemic [21]. Two weeks after the second CoronaVac dose, we detected seroconversion for anti-S1 IgG in 98.0% of vaccinees, with a median titer of 206.7 BAU/mL. This finding conforms to previous studies, reporting rates between 97.0% and 100% (Brazil [19,22]), China [19], Turkey [23]) and median anti-S1 IgG titers of 106 to 429 BAU/mL [14,19] within 2–3 weeks after the second CoronaVac dose. In contrast, others found anti-S1 IgG positivity in only 75.0–77.4% (Chile) [16,24]. The relevance of the second vaccine dose for eliciting markedly increased antibody titers has also been established for other COVID-19 vaccination regimens [25,26,27]. Compared to CoronaVac, both vector- and mRNA-based vaccines (e.g., ChAdOx1 and BNT162b2) elicit much higher anti-S1 titers and positivity rates already after the first vaccine dose. After the second dose, these responses continue to rise, leading to very high seropositivity rates, also among CoronaVac vaccinees (Appendix A) [19,27]. For example, Lau et al. compared early immune responses to BNT162b2 with those following CoronaVac among Singaporean vaccinees. Within 10–20 days after the first/second/third dose, the median levels of total spike antibodies elicited by BNT162b2 (2.48/2174/15,004 BAU/mL) were significantly higher than those elicited by CoronaVac (0.4/98/525 BAU/mL), corresponding to 6.2/22.2/28.6-fold higher responses in the mRNA vaccine recipients. For both vaccine types, there were no significant differences in antibody levels between male and female vaccinees at all time points, which is consistent with our study [28]. Another Singaporean study also revealed higher antibody responses to the mRNA vs. inactivated vaccine for both total spike Ig (3.29 vs. 0.39/2219 vs. 97.0/19,562 vs. 555 BAU/mL) and anti-spike IgG (17.8 vs. 0.50/2271 vs. 106/2932 vs. 131 BAU/mL) within 10–20 days after the first/second/third dose [14]. Valyi-Nagy et al. studied responses to BBIBP-CorV (Sinopharm, Beijing, China), another inactivated COVID-19 vaccine, and found that the levels of anti-S1/S2 antibodies were 6.4-fold higher (median 517.8 AU/mL) in BNT162b2 vaccinees than in BBIBP-CorV vaccinees (median 80.5 AU/mL) at 1–2 weeks after the second dose [29].

We detected anti-S1 IgA in the plasma of 78.0% cases, presenting a higher rate than reported for Brazilian vaccinees (56.0%) 2 weeks after the second CoronaVac dose [19]. Our finding of two vaccinees with borderline anti-S1 IgA levels prior to vaccination may be related to pre-existing cross-reactive antibodies resulting from previous exposure to other human coronaviruses. There is growing evidence supporting the clinical importance of measuring spike-specific serum IgA in patients with suspected or confirmed COVID-19. As such, the results may (i) contribute to diagnosing SARS-CoV-2 infections if molecular testing yields negative/inconclusive results, (ii) enhance the accuracy of serological assessment, and (iii) predict disease severity and progression [30,31,32,33]. Pisanic et al. demonstrated high correlations between serum and saliva levels of anti-SARS-CoV-2 IgA, suggesting that serum IgA titers reflect the mucosal immunity status [34]. However, this contrasts with findings by Chan et al., who detected significant increases in anti-S1 IgA, IgG and neutralization activity in plasma samples after CoronaVac vaccination, while they were unable to detect these antibodies in nasal mucosa fluid [35].

Interestingly, CoronaVac elicited a significant increase in anti-NCP IgG, but the median antibody level at t2 (ratio 0.6, 6-fold increase from t0 to t2) remained below the assay cut-off and was distinctly inferior to the S1-specific responses at t2 (206.7 BAU/mL, 65-fold increase from t0 to t2). This is consistent with studies on the immunogenicity of CoronaVac in mice and rats, where the amount of anti-NCP IgG was 30-fold lower than that of spike- and S1/RBD-specific IgG, suggesting that the latter represent the dominant antigens [36]. Similarly, Bueno et al. reported fold changes for NCP-specific IgG in Chilean adults of 1.61 and 1.89 (2 and 4 weeks after the second CoronaVac dose, respectively), while anti-S1/RBD IgG increased with fold changes ranging from 29.72 to 31.15 [24]. We found seroconversion against NCP in 40.0% of cases at t2, which is in the range of rates reported for Brazilian (51.9–64.0%) [19,22], Hong Kong Chinese (59.2%) [37] and Chilean vaccinees 6.9% [24] 2–4 weeks after the second CoronaVac dose. Among Hungarian BBIBP-CorV recipients, 35.0% seroconverted for anti-NCP IgG [29]. Similarly to our study, Hayashi et al. used the EUROIMMUN assay for anti-NCP IgG testing in a Brazilian group of adult CoronaVac vaccinees and found median ratios ≤0.2 at t0 and t1 (before and 28 days after first dose), followed by an increase to ratio 1.6 at t2 (14 days after second dose; negative cut-off ratio 0.8) and a seroconversion rate of 64.0% [19]. In another study, anti-NCP IgG measurement among Hong Kong Chinese adults revealed a seroconversion rate of 59.2% 4 weeks after the second CoronaVac dose and a median OD_450_ value of 1.0, which was only slightly above the negative cut-off value (OD_450_ 0.9) of their assay [37].

During natural SARS-CoV-2 infection, seroconversion for anti-S1 IgG and anti-NCP total Ig was observed in 92.5% and 98.8% of Turkish patients 6 weeks after hospital admission, whereas CoronaVac-induced rates in a matched comparison group amounted to 96.2% and 51.2% after the second dose, respectively. Antibody titers against NCP were shown to be significantly lower (median 0.992) in CoronaVac recipients than after natural infection (median 34.95), with significant differences between patients with non-severe (median 26.02) and severe COVID-19 (median 63.14). In contrast, CoronaVac-induced anti-S1 titers (median 3.17) were significantly lower than in infected patients displaying severe symptoms (median 5.952), while there was no significant difference to those with non-severe disease [38].

Reasons explaining the lower median titers of both anti-S1 and anti-NCP antibodies in CoronaVac responses compared to (severe) COVID-19 cases may include the lower amount of virus in the vaccine, while high viral loads in severely diseased patients cause a stronger increase in post-infection antibody levels [38]. Secondly, antigen multivalency and alterations in spike protein due to chemical inactivation, which are common characteristics of inactivated viral vaccines, could result in inferior immunogenicity compared to natural infection and S-based vaccine types [39]. Thirdly, antibody avidity might have an influence on detection levels. Spike-specific IgG avidity was shown to be significantly lower in CoronaVac recipients than in BNT162b2 recipients or COVID-16 convalescent patients [37], and anti-NCP antibodies are of lower avidity than spike-specific antibodies [40]. This, together with the fact that indirect ELISA formats are capable of detecting high-avidity antibodies more effectively than low-avidity antibodies [40], may contribute to the measurement of low antibody titers.

In the present study, the quantitative anti-NCP IgG values positively correlated with anti-S1 IgG (r_s_ = 0.624, *p* < 0.001) and anti-S1 IgA (r_s_ = 0.527, *p* < 0.001). The highest correlation was observed between anti-S1 IgG and anti-S1 IgA (r_s_ = 0.767, *p* < 0.001), possibly reflecting the above-mentioned disparity in NCP- and S-based responses to CoronaVac. Positive correlations between NCP- and spike-specific antibodies (r > 0.6, *p* < 0.001) have also been reported following SARS-CoV-2 infection [41,42] and vaccination with BNT162b2 [43].

Of note, neither NCP- nor spike-based serology are appropriate to distinguish natural (and breakthrough) infections from vaccination response in populations using CoronaVac for immunization. This is in contrast to the situation in populations using spike vaccines, allowing the application of anti-NCP-based serology to identify antibodies related to wildtype virus infection [44].

The limitations to this study include, first, the small number of participants and a lack of at least one matched comparison group that received another vaccine. Second, the study period was short and without follow-up, preventing inferences on the longevity of humoral responses. However, previous studies demonstrated a peak in antibody levels 20–30 days after the second CoronaVac dose, followed by a steady decrease [14,15,16,22,27]. Third, the data do not allow for conclusions on whether the detected antibodies contribute to protective immunity because neither neutralization activity nor cellular responses were tested. However, CoronaVac’s ability to induce humoral responses with neutralizing capacity as well as the correlation between antibody titers and neutralizing effects have already been described elsewhere [14,35,45,46,47].

## 5. Conclusions

Two doses of the whole-virion CoronaVac vaccine are capable of inducing a significant increase in NCP- and S1-specific antibodies, indicating successful stimulation of the humoral response against SARS-CoV-2 among adults aged 27 to 58 years. Despite strong correlations between the antibody concentrations, the median levels and seroconversion rates of anti-S1 IgG and IgA exceed those of anti-NCP IgG.

## Figures and Tables

**Figure 1 vaccines-11-00216-f001:**
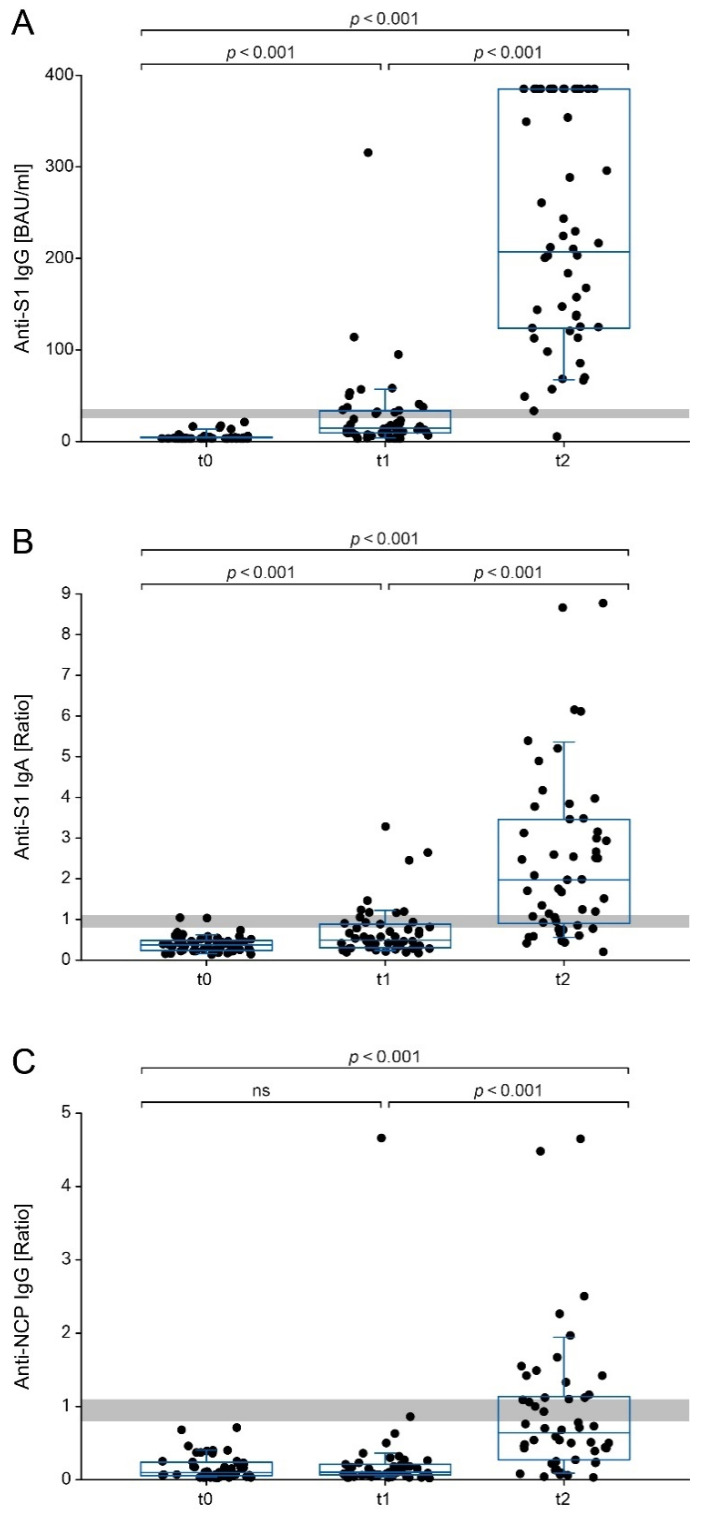
(**A**) Anti-S1 IgG, (**B**) anti-S1 IgA and (**C**) anti-NCP IgG reactivity in 50 vaccinees at baseline (t0), 4 weeks after the first dose (t1), and 2 weeks after the second dose of CoronaVac (t2). Data are presented as scatter plots, boxes indicate interquartile ranges (outer bounds) and medians (midlines). Whiskers present the 90th and 10th percentiles. Borderline ranges are shaded in grey. The Wilcoxon signed-rank test was used to analyze differences in antibody levels between the time points (ns, not statistically significant). For anti-S1 IgG, samples with results above the measurement range (>384 BAU/mL) are indicated as data points at 385 BAU/mL.

**Figure 2 vaccines-11-00216-f002:**
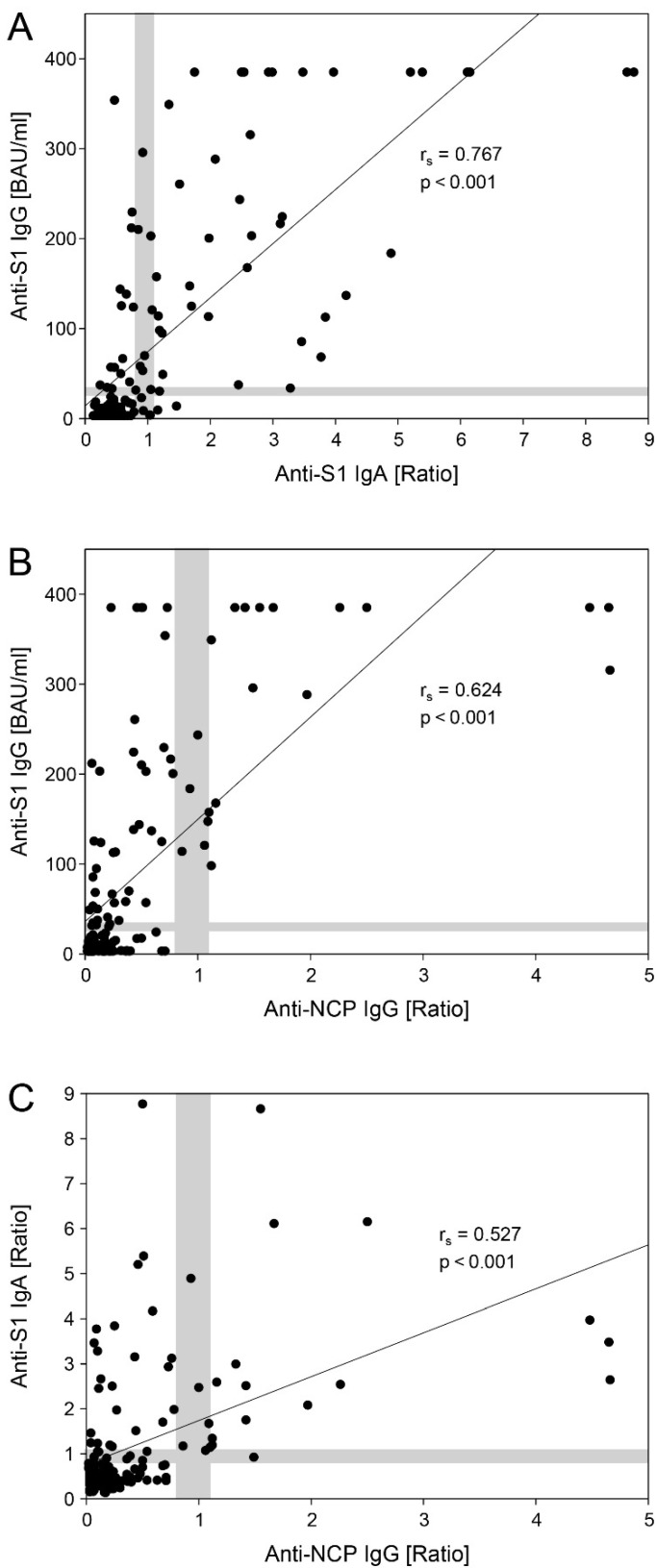
Correlation between anti-S1 IgG, anti-S1 IgA and anti-NCP IgG reactivity in a total of 150 samples collected from 50 vaccinees at baseline (t0), 4 weeks after the first dose (t1), and 2 weeks after the second dose of CoronaVac (t2). (**A**) Anti-S1 IgG vs. anti-S1 IgA, (**B**) anti-S1 IgG vs. anti-NCP IgG, (**C**) anti-S1 IgA vs. anti-NCP IgG. Borderline ranges are shaded in grey; r_s_, Spearman correlation coefficient. For anti-S1 IgG, samples with results above the measurement range (>384 BAU/mL) are indicated as data points at 385 BAU/mL.

**Table 1 vaccines-11-00216-t001:** Median titers and positivity rates of SARS-CoV-2-specific antibodies in 50 Turkish adults before and after administration of two doses of CoronaVac vaccine.

Sampling Time	Anti-S1 IgG			Anti-S1 IgA			Anti-NCP IgG		
	Median [Range]	Rate (n/N) ^a^	95% CI	Median [Range]	Rate (n/N) ^a^	95% CI	Median [Range]	Rate (n/N) ^a^	95% CI
t0 (baseline pre-vaccination)	<3.2 [<3.2–21.1] BAU/mL	0% (0/50)	0–8.5%	Ratio 0.4 [0.1–1.0]	4.0% (2/50)	0.3–14.2%	Ratio 0.1 [0.0–0.7]	0% (0/50)	0–8.5%
t1 (28–29 days after first dose)	13.8 [<3.2–315.4] BAU/mL	30.0% (15/50)	19.0–43.8%	Ratio 0.5 [0.2–3.3]	28.0% (14/50)	17.4–41.8%	Ratio 0.1 [0.0–4.7]	4.0% (2/50)	0.3–14.2%
t2 (14–15 days after second dose)	206.7 [5.3–384.0] BAU/mL	98.0% (49/50)	88.5–99.9%	Ratio 2.0 [0.2–8.8]	78.0% (39/50)	64.6–87.4%	Ratio 0.6 [0.0–4.7]	40.0% (20/50)	27.6–53.8%

Abbreviations: BAU/mL, binding antibody units per milliliter; CI, confidence interval; ELISA, enzyme-linked immunosorbent assay; NCP, nucleocapsid protein; SARS-CoV-2, severe acute respiratory syndrome-associated coronavirus 2; S1, spike protein subunit S1. ^a^ Borderline results were considered positive.

## Data Availability

The data presented in this study are available on request from the corresponding author.

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
