# Peer review of "Humoral Immune Response to CoronaVac in Turkish Adults"

_vaccines, 2023, doi:10.3390/vaccines11020216_

Round 1

Reviewer 1 Report

There is a paper with similar design, timing but involves 3 doses of Sinovac & compared to Pfizer mRNA vaccine which you might want to review & compare - Lau CS et al. Vaccines 2022;11(2):38

1. What is the main question addressed by the research? The literature & interest around the humoral response to 2 doses of Sinovac would have been interesting in 2021 when vaccines were scarce especially as there was little published data on CoronaVac. Now we know that the primary series with CoronaVac should be 3 or even 4 doses. Besides early antibody kinetics 2 weeks after dose 2 is of little value as that is the expected peak antibody response. We also know that antibodies wane quickly - .>50% by day 40 after dose 2 (Antibodies Lau 2022;11:70). There is also no comparison with other non-CoronaVac vaccines
2. Do you consider the topic original or relevant in the field?  NO.

Does it address a specific gap in the field? NO.
3. What does it add to the subject area compared with other published material?

Very little other than data from Turkey & studies using serology from Euroimmun platform.

4. What specific improvements should the authors consider regarding the methodology? 

Longer time points to emphasise the inadequacy of 2 CoronaVac shots & Comparison with other non-CoronaVac vaccines.

What further controls should be considered?

While the IgA kinetics might be some new information, it is unlikely to be a significant marker given the underwhelming IgA response compared to IgG and its more rapid decline. Perhaps total SARS-CoV-2 spike-Ab might be more useful.  Measurement of neutralizing antibodies would have enhanced the paper BUT it should be mentioned as a limitation in the discussion section.
5. Are the conclusions consistent with the evidence and arguments presented and do they address the main question posed?
Yes but insufficient.

6. Are the references appropriate?

Yes but greater in-depth comparisons with other studies at similar vaccine time points might have been helpful to determine if there are regional/ethnic differences between Turkish residents and other elsewhere.

7. Please include any additional comments on the tables and figures. Table 1 should include not only the median antibody levels but also the range of values found.   Though well written, I am now inclined to recommend that Vaccines reject this paper & encourage re-submission after they address the shortcomings highlighted.

Reviewer 2 Report

The design of the study was very well done. The manuscript brings relevant datas that must be publish. The results clearly show that CoronaVac vaccine is able to induce a robust humoral Immune response to the S1 and NCP antigens of the Sars-CoV-2 after the boost second dose. 

I would like to suggest the authors expand the introduction bringing other studies that report success of the CoronaVac vaccination in other population groups. As suggestion I propose the following:

Ranzani OT, Hitchings MDT, Dorion M, D'Agostini TL, de Paula RC, de Paula OFP, Villela EFM, Torres MSS, de Oliveira SB, Schulz W, Almiron M, Said R, de Oliveira RD, Vieira da Silva P, de Araújo WN, Gorinchteyn JC, Andrews JR, Cummings DAT, Ko AI, Croda J. Effectiveness of the CoronaVac vaccine in older adults during a gamma variant associated epidemic of covid-19 in Brazil: test negative case-control study. BMJ. 2021 Aug 20;374:n2015. doi: 10.1136/bmj.n2015. Erratum in: BMJ. 2021 Sep 6;374:n2091. PMID: 34417194; PMCID: PMC8377801.

Florentino PTV, Alves FJO, Cerqueira-Silva T, Oliveira VA, Júnior JBS, Jantsch AG, Penna GO, Boaventura V, Werneck GL, Rodrigues LC, Pearce N, Barral-Netto M, Barreto ML, Paixão ES. Vaccine effectiveness of CoronaVac against COVID-19 among children in Brazil during the Omicron period. Nat Commun. 2022 Aug 13;13(1):4756. doi: 10.1038/s41467-022-32524-5. PMID: 35963844; PMCID: PMC9375192.

Bichara CDA, Queiroz MAF, da Silva Graça Amoras E, Vaz GL, Vallinoto IMVC, Bichara CNC, Amaral IPCD, Ishak R, Vallinoto ACR. Assessment of Anti-SARS-CoV-2 Antibodies Post-Coronavac Vaccination in the Amazon Region of Brazil. Vaccines (Basel). 2021 Oct 12;9(10):1169. doi: 10.3390/vaccines9101169. PMID: 34696277; PMCID: PMC8539673.

The last one manuscript is interesting be cited because it shows similar levels of antibody response among young adults, wich decrease among elder.

Additionally, considering that the authors analyzed individuals aged between 27 and 54 years old, I suggest that the conclusion highlight the success of the vaccine to this age group. 

Reviewer 3 Report

In the manuscript titled “Humoral immune response after immunization with inactivated whole-virion SARS-CoV-2 vaccine (CoronaVac) in Turkish adults”, Cosgun et al. present results of short-term humoral (anti-S1 IgG and IgA, anti-NCP IgG) responses against SARS-CoV-2 in patients who received CoronaVac vaccine.

The results are presented clearly, and the paper is well-written. However, the major drawback is originality since humoral immune response in vaccinated individuals was already presented in many studies and is not representative of true vaccine protection qualities. The limitations include the small number of participants and the short-term observation period, excluding the effects of a possible third dose. However, the results are well presented and can be considered an additional piece of the anti-SARS-CoV-2 immune response puzzle. The significance of the results can also be attributed to a substantial proportion of people vaccinated with CoronaVac worldwide.

Specific points to be regarded:

  • The title can be simplified.
  • Although the figures are relatively easy to understand, the legends should include separate references to A, B and C sections to clarify them.
  • The authors should expand the Discussion with more data on NCP-specific humoral response during natural infection and possible explanations for its absence or low levels in patients who received this and other whole-virion vaccines.
  • Some comments can be added on the correlation between anti-S1 IgG, IgA, and anti-NCP IgG levels.
